# Synaptic location is a determinant of the detrimental effects of α-synuclein pathology to glutamatergic transmission in the basolateral amygdala

Liqiang Chen[1,2†], Chetan Nagaraja[1†], Samuel Daniels[1†], Zoe A Fisk[2,3,4,5], Rachel Dvorak[1], Lindsay Meyerdirk[1,2], Jennifer A Steiner[1], Martha L Escobar Galvis[1], Michael X Henderson[1,2], Maxime WC Rousseaux[2,3,4,5], Patrik Brundin[6], Hong-Yuan Chu[1,2]*

[1]Department of Neurodegenerative Science, Van Andel Institute, Grand Rapids, United States; [2]Aligning Science Across Parkinson's (ASAP) Collaborative Research Network, Chevy Chase, United States; [3]Department of Cellular and Molecular Medicine, University of Ottawa, Ottawa, Canada; [4]University of Ottawa Brain and Mind Research Institute, Ottawa, Canada; [5]Ottawa Institute of Systems Biology, Ottawa, Canada; [6]Pharma Research and Early Development (pRED), F. Hoffman-La Roche, Little Falls, United States

**\*For correspondence:**
hongyuan.chu@vai.org

[†]These authors contributed equally to this work

**Abstract** The presynaptic protein α-synuclein (αSyn) has been suggested to be involved in the pathogenesis of Parkinson's disease (PD). In PD, the amygdala is prone to develop insoluble αSyn aggregates, and it has been suggested that circuit dysfunction involving the amygdala contributes to the psychiatric symptoms. Yet, how αSyn aggregates affect amygdala function is unknown. In this study, we examined αSyn in glutamatergic axon terminals and the impact of its aggregation on glutamatergic transmission in the basolateral amygdala (BLA). We found that αSyn is primarily present in the vesicular glutamate transporter 1-expressing (vGluT1[+]) terminals in the mouse BLA, which is consistent with higher levels of αSyn expression in vGluT1[+] glutamatergic neurons in the cerebral cortex relative to the vGluT2[+] glutamatergic neurons in the thalamus. We found that αSyn aggregation selectively decreased the cortico-BLA, but not the thalamo-BLA, transmission; and that cortico-BLA synapses displayed enhanced short-term depression upon repetitive stimulation. In addition, using confocal microscopy, we found that vGluT1[+] axon terminals exhibited decreased levels of soluble αSyn, which suggests that lower levels of soluble αSyn might underlie the enhanced short-term depression of cortico-BLA synapses. In agreement with this idea, we found that cortico-BLA synaptic depression was also enhanced in αSyn knockout mice. In conclusion, both basal and dynamic cortico-BLA transmission were disrupted by abnormal aggregation of αSyn and these changes might be relevant to the perturbed cortical control of the amygdala that has been suggested to play a role in psychiatric symptoms in PD.

## Editor's evaluation

The manuscript by Chen et al., examines the synapse-specificity of α-synuclein aggregation and corresponding circuit dysfunction in the amygdala. Using confocal microscopy and slice electrophysiology, along with α-synuclein knockout mice and preformed fibrils, the authors demonstrate that cortico-amygdala, but not thalamo-amygdala, inputs are more vulnerable to α-synuclein aggregation

and corresponding synaptic dysfunction. This has important implications for the etiology of psychiatric deficits that are common in Parkinson's disease.

## Introduction

α-synuclein (αSyn) is a soluble protein abundant at presynaptic axon terminals, where it regulates the dynamics of synaptic vesicles through interaction with synaptic proteins and presynaptic membranes (*Burré et al., 2010*; *Runwal and Edwards, 2021*; *Vargas et al., 2017*). αSyn is also prone to form insoluble cytoplasmic aggregates, which are the major protein component of Lewy pathology seen in synucleinopathies like Parkinson's disease (PD) (*Mezey et al., 1998*; *Spillantini et al., 1997*). Increasing evidence supports the notion that pathologic αSyn propagates between synaptically interconnected brain regions and underlies PD progression (*Angot et al., 2010*; *Luk et al., 2012*; *Uemura et al., 2020*).

The amygdala is a key limbic structure for emotion regulation (*Janak and Tye, 2015*). Compelling clinical evidence indicates that cortical control of the amygdala activity is impaired in PD, leading to an inappropriate encoding of emotion valence or deficits in linking emotion to behavior (*Bowers et al., 2006*; *Hu et al., 2015*; *Yoshimura et al., 2005*). Moreover, the amygdala shows selective vulnerability to Lewy pathology (*Harding et al., 2002*; *Nelson et al., 2018*; *Sorrentino et al., 2019*), thus an impaired amygdala network function has been proposed to underlie the disrupted emotion processing in PD patients (*Harding et al., 2002*). Still, the normal function of αSyn and how its aggregation can impair amygdala circuit function remain poorly understood. Here, we show that αSyn is primarily present in vesicular glutamate transporter 1-expressing (vGluT1⁺) cortical axon terminals, and, by contrast, is barely detectable in vGluT2⁺ thalamic axon terminals in the basolateral amygdala (BLA). In an αSyn preformed fibrils (PFFs) model of synucleinopathies, αSyn pathology decreases vGluT1⁺ cortico-BLA glutamatergic transmission, without affecting the vGluT2⁺ thalamo-BLA neurotransmission. Furthermore, we demonstrate that a partial (secondary to αSyn aggregation) or complete (*Snca* KO mice) depletion of soluble αSyn from the axon boutons promotes short-term depression at cortico-BLA synapses in response to prolonged stimulation, leading to an impaired gain control of cortical inputs to the BLA. Therefore, we conclude that both gains of toxic properties and loss of normal function of αSyn contribute to the input-specific disruption of cortico-BLA synaptic connectivity and plasticity in synucleinopathies. Our data support clinical observations of an impaired cortical control of the amygdala activity that could contribute to psychiatric deficits in PD patients.

## Results

### αSyn localizes preferentially in vGluT1⁺ axon terminals in mouse brain

Compelling evidence from PD patients suggests that the amygdala exhibits an impaired responsiveness to sensory stimuli, arising mainly from the cerebral cortex and the thalamus (*Bowers et al., 2006*; *Hu et al., 2015*). Thus, we examined the presence of αSyn in vGluT1⁺ and vGluT2⁺ axon boutons in wild type (WT) mouse brains, which mainly come from cortical and thalamic regions, respectively (*Fremeau et al., 2001*; *Kaneko and Fujiyama, 2002*; *Vigneault et al., 2015*). *Figure 1* shows that αSyn immunoreactivity colocalizes with vGluT1⁺ puncta but is absent where there are vGluT2⁺ puncta, in the BLA, the cerebral cortex, and the striatum.

Earlier in situ hybridization studies showed higher αSyn mRNA expression in the cerebral cortex and the hippocampus than in the thalamus (*Abeliovich et al., 2000*; *Ziolkowska et al., 2005*). Next, we used *Snca^{NLS/NLS}* reporter mice to determine αSyn protein localization in the cortical and thalamic areas. These mice localize endogenous αSyn to the nucleus that allows the visualization of cellular topography (*Geertsma et al., 2022*), circumventing the diffused αSyn immunoreactivity in WT mice (*Figure 2A*). We observed that the αSyn was heavily expressed in cortical layer V/VI neurons, but only moderately or barely expressed in thalamic regions, particularly in the midline thalamus that provides major excitation to the BLA (*Ahmed et al., 2021*; *Amir et al., 2019*; *Hintiryan et al., 2021*; *Figure 2B–D*). Together, our data show that αSyn is preferentially present at vGluT1⁺ cerebral cortical neurons and their axon terminals but is absent or expressed at very low levels at vGluT2⁺ thalamic neurons and their projections.

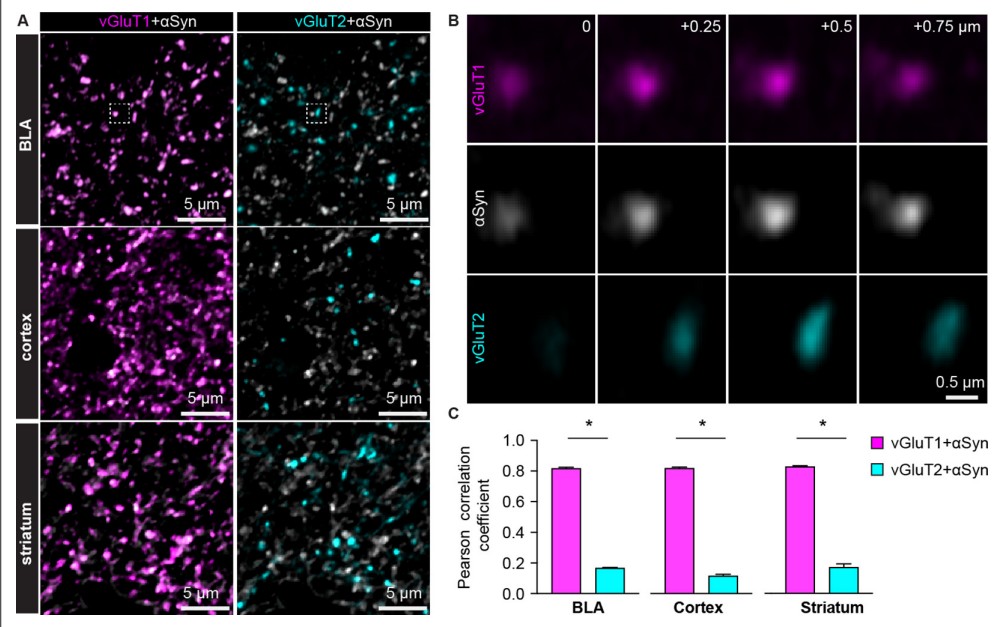

**Figure 1.** α-synuclein (αSyn) is selectively localized at vesicular glutamate transporter 1 (vGluT1[+]) axon terminals in mouse brain. (**A**) Representative confocal images showing colocalization of αSyn with vGluT1 (left), but not with vGluT2 (right) in the basolateral amygdala (BLA) (top), cerebral cortex (middle), and the striatum (bottom). Simultaneously collected confocal images from each brain region were split into vGluT1/αSyn and vGluT2/αSyn channels for the purpose of illustration. (**B**) Zoomed z series of images of the boxed area from the BLA in (**A**) showing colocalization and correlated changes in the immunoreactive intensities of vGluT1 and αSyn. Such colocalization and correlation are absent between vGluT2 and αSyn within the same region. Images were taken and shown from a + μm z-depth with an inter-section interval of 0.25 μm. (**C**) Bar graphs showing Pearson correlation coefficient between vGluT1 and αSyn, as well as between vGluT2 and αSyn, in the BLA (αSyn/vGluT1=0.82 ± 0.007, αSyn/vGluT2=0.17 ± 0.004; n=18 slices/4 mice; p<0.0001, MWU), cerebral cortex (αSyn/vGluT1=0.82 ± 0.006, αSyn/vGluT2=0.12 ± 0.009; n=8 slices/3 mice; p=0.0002, MWU), and the striatum (αSyn/vGluT1=0.83 ± 0.006, αSyn/vGluT2=0.17 ± 0.02; n=8 slices/3 mice; p=0.0002, MWU).

The online version of this article includes the following source data for figure 1:

**Source data 1.** Source data for plot in *Figure 1C*.

## Glutamate release from vGluT1[+] axon terminals is selectively disrupted by αSyn pathology

Considering that the endogenous levels of αSyn are determinants of the propensity to form αSyn aggregates, we hypothesize that vGluT1[+] neurons and their terminals are more susceptible to αSyn pathology compared to those are vGluT2[+] (*Erskine et al., 2018*; *Vasili et al., 2022*). To test this hypothesis, we triggered widespread αSyn pathology in the brain using the intrastriatal PFFs seeding model (*Luk et al., 2012*). One-month post-injection, we detected robust αSyn pathology in vGluT1[+] cerebral cortical regions (e.g. the temporal association cortex (TeA), the motor cortex, and the perirhinal cortex, *Figure 3 and A1–A*) and the BLA (*Figure 3B–C*), but we barely observed any cytoplasmic aggregates in vGluT2[+] thalamic regions (*Figure 3 and A4–A5*). This pattern of pathology is supported by differences in endogenous αSyn levels between the cerebral cortex and thalamus (*Figure 2B–D*) and is consistent with earlier reports (*Burtscher et al., 2019*; *Henderson et al., 2019*; *Luk et al., 2012*; *Stoyka et al., 2020*). Importantly, neurons in the midline thalamus were retrogradely labeled when retrobeads were injected into the same location as for PFFs in the striatum (*Figure 3A6*). We concluded that the absence of cytoplasmic pS129 αSyn pathology in the thalamus was not due to technical issues (i.e. missing thalamostriatal axon terminals for PFFs internalization).

Next, we focused on the BLA to assess the functional impact of αSyn aggregation on glutamatergic transmission from the cerebral cortex and thalamus (*Abeliovich et al., 2000*; *Ziolkowska et al., 2005*). We selectively activated cortical and thalamic inputs of the BLA by stimulating the external and internal capsules, respectively. One-month post-injection, the amplitude of electrically evoked

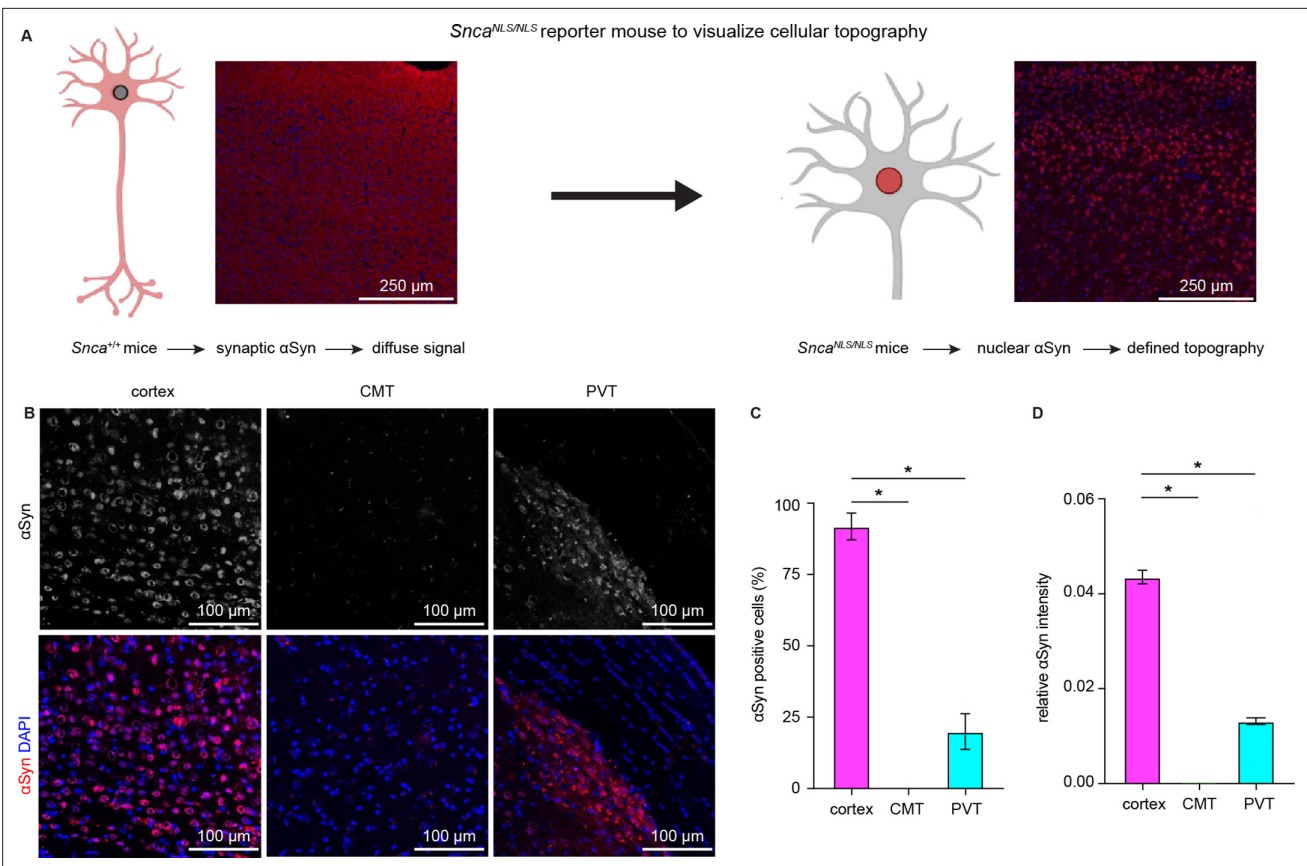

**Figure 2.** Cerebral cortical neurons express higher levels of endogenous α-synuclein (αSyn) than thalamic neurons. (**A**) Schematic of approach to determine cellular topography of αSyn in the *Snca*<sup>NLS/NLS</sup> reporter mouse line. (**B**) Representative photomicrographs in the different highlighted brain regions for either αSyn alone (top panels) or co-stained with DAPI as a nuclear marker. (**C–D**) Quantification of the proportion of αSyn-positive nuclei (C, % αSyn positive cells, cortex=91.9 ± 4.7%, CMT=0 ± 0%; PVT=19.9 ± 6.3%, n=3 mice) or average relative intensity of αSyn in the different regions (cerebral cortex=0.04 ± 0.001, n=742 cells/3 mice; CMT=0 ± 0, n=364 cells/3 mice; PVT=0.01 ± 0.0007, n=734 cells/3 mice). * p<0.05, one-way ANOVA followed by Sidak's multiple comparison tests. Abbreviations: CMT, centromedial thalamus; PVT, periventricular thalamus.

The online version of this article includes the following source data for figure 2:

**Source data 1.** Source data for plots in *Figure 2C and D*.

cortico-BLA excitatory postsynaptic currents (EPSCs) was greatly decreased in the slices from PFFs-injected WT mice relative to those from controls (*Figure 3D–E*). Consistently, prominent pS129 αSyn pathology can be detected along the external capsule where the cortical afferents enter the BLA (*Figure 3B–C*). In contrast, there was no difference in the amplitude of thalamo-BLA EPSCs between the PFFs-injected mice and controls (*Figure 3F–G*).

To avoid potential technical issues inherent to the use of electrical stimulation, we employed opto-genetic approach to confirm the above results (*Figure 3H–O*). One-month post-injection, the amplitude of optogenetically-evoked cortico-BLA EPSCs in slices from PFFs-injected mice was decreased relative to those from PBS-injected controls (*Figure 3H–K*). By contrast, we did not detect the difference in the amplitude of optogenetically-evoked thalamo-BLA EPSCs between groups (*Figure 3L–O*). Altogether, we demonstrated that the presence of αSyn makes vGluT1⁺ cortical neurons and their axons more vulnerable to αSyn pathology and that this pathology is associated with detrimental effects on synaptic transmission.

Moreover, consistent with the development of αSyn pathology requiring the presence of endog-enous αSyn in the PFFs model (*Luk et al., 2012*; *Volpicelli-Daley et al., 2011*), we did not detect the difference in the cortico-BLA transmission between PFFs- versus PBS-injected αSyn KO mice (*Figure 3—figure supplement 1*). Furthermore, because of the lack of detrimental effect in αSyn KO mice, we conclude that the impaired cortico-BLA transmission in the PFFs model is likely caused

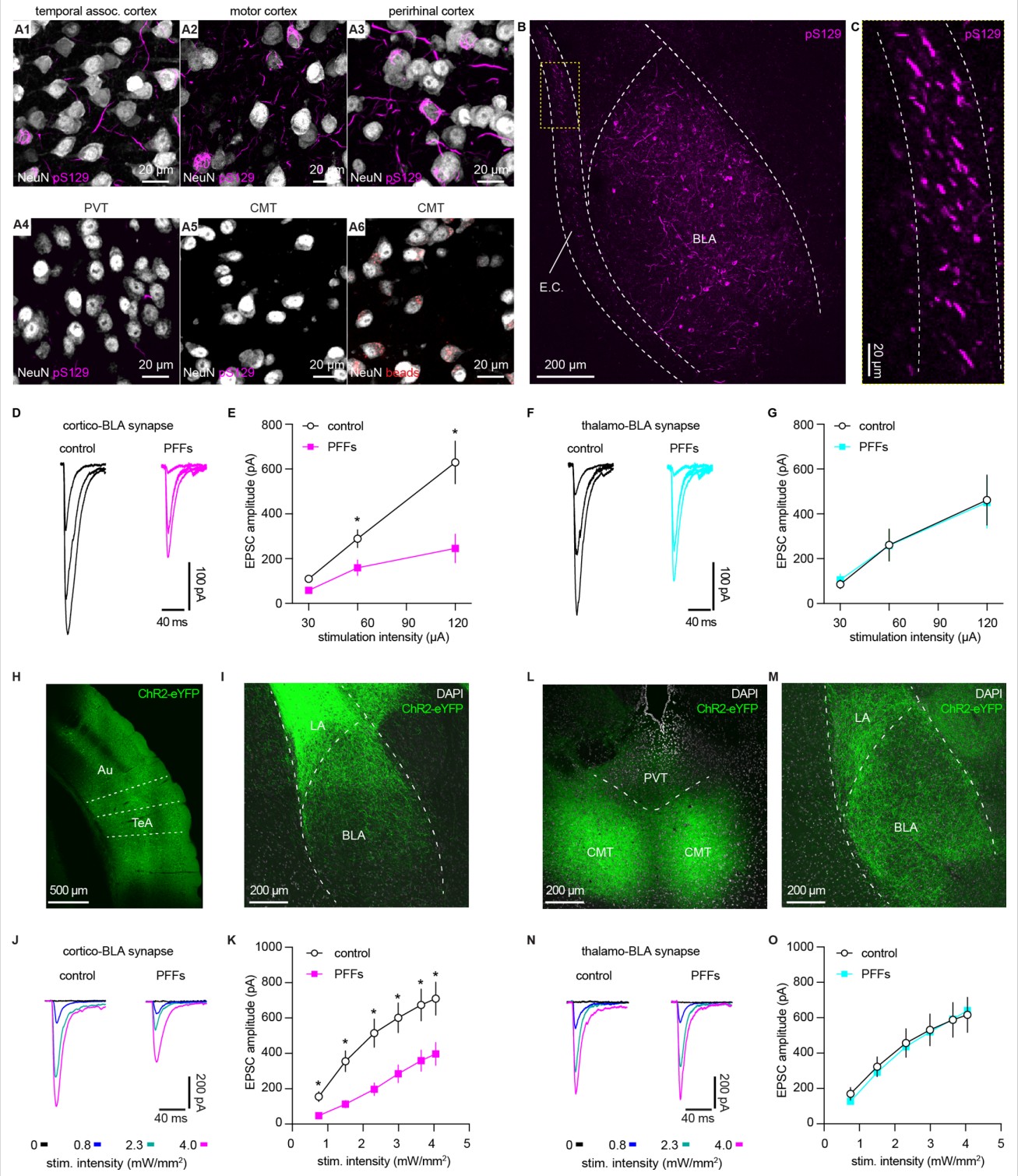

**Figure 3.** α-synuclein (αSyn) aggregates preferentially disrupt vesicular glutamate transporter 1 (vGluT1⁺) cortico-BLA transmission. (**A**) Representative images showing the presence of αSyn aggregates in the cortical regions (**A1–A3**), but largely absent in the midline thalamus (**A4–A5**). A6, representative images showing retrobeads labeled neurons in the centromedial thalamus. (**B–C**) Representative images showing pS129⁺ αSyn pathology in the basolateral amygdala (BLA) (**B**) and the external capsule (**C**). (**D–E**) Representative traces of excitatory postsynaptic currents (EPSCs) evoked by electrical stimulation of the external capsule (**D**) and summarized results (**E**) showing a reduced cortico-BLA transmission in slices from preformed fibrils (PFFs)- versus PBS-injected mice. n=17 neurons/4 mice for each group. (**F–G**) Representative traces of EPSCs evoked by electrical stimulation of the internal capsule (**F**) and summarized results (**G**) showing unaltered thalamo-BLA transmission in slices from PFFs- versus PBS-injected wildtype mice.

*Figure 3 continued on next page*

*Figure 3 continued*

n=17 cells/5 mice for controls, and 13 cells/4 PFF-injected mice. (**H–I**) Representative images showing viral infection site in the temporal association cortex (TeA) and nearby regions (**H**), and the axon terminal field in the BLA (**I**). (**J–K**) Representative traces of optogenetically-evoked EPSCs (**J**) and summarized results (**K**) showing a reduced amplitude of cortico-BLA EPSCs in slices from PFFs- versus PBS-injected mice. n=35–37 neurons/4 mice per group. (**L–M**) Representative images showing viral infection site in the midline thalamus (**L**), and the axon terminal field in the BLA (**M**). (**N–O**) Representative traces of optogenetically-evoked EPSCs (**N**) and summarized results (**O**) showing unaltered thalamo-BLA transmission in slices from PFFs- versus PBS-injected mice. n=28 neurons/4 mice per group. *, p<0.05, MWU followed by Bonferroni-Dunn correction for multiple comparisons. Abbreviations: CMT, centromedial thalamus; PVT, periventricular thalamus.

The online version of this article includes the following source data and figure supplement(s) for figure 3:

**Source data 1.** Source data for plots in *Figure 3*.

**Figure supplement 1.** Intrastriatal preformed fibrils (PFFs) injection does not alter cortico-BLA transmission in synKO mice.

**Figure supplement 1—source data 1.** Source data for plot in *Figure 3—figure supplement 1*.

by the gained toxic properties of αSyn as it aggregates, instead of intrastriatal PFFs injection per se (*Cookson and van der Brug, 2008*; *Volpicelli-Daley et al., 2011*).

## αSyn pathology decreases the number of functional cortico-BLA inputs

Several mechanisms can contribute to the decreased cortico-BLA synaptic strength as αSyn pathology develops, including the loss of synapses, decreased initial release probability, and/or postsynaptic adaptations. To determine the impact of αSyn aggregation on vGluT1$^+$ cortico-BLA innervation, brain sections from control and PFFs-injected mice were processed for immunohistochemical assessment of vGluT1. The density of vGluT1-immunoreactive puncta in the BLA was then determined stereologically (*West, 1999*). The density of vGluT1-immunoreactive puncta was not altered between PFFs- and PBS-injected mice (*Figure 4A–C*). These results indicate that there is no loss of cortical axon terminals or cortico-BLA synapses in PFFs-injected mice at one-month post-injection.

Next, we assessed functional changes in cortico-BLA synaptic transmission using electrophysiology. The initial release probability was estimated by delivering paired-pulses of electric or optogenetic stimulation of cortico-BLA synapses in control and PFFs-injected mice. Using electric stimulation approach, the ratio of EPSC2 to EPSC1 (at a 50 ms inter-pulse interval) was not altered between groups (*Figure 4D–E*). Similarly, the ratio of EPSC2/EPSC1 at cortico-BLA synapses was not altered by αSyn pathology when assessed using optogenetics (at a 100 ms inter-pulse interval, *Figure 4F–G*). Interestingly, in contrast to the short-term facilitation in response to electric stimulation (*Figure 4D–E*), paired-pulses of optogenetic stimulation always led to short-term depression of cortico-BLA transmission (*Figure 4F–G*), which might be due to the deactivation of ChR2 itself. Together, the above data suggest that the decreased cortico-BLA transmission was not caused by a lower initial release probability.

Further, we estimated the quantal properties of cortico-BLA synapses by measuring the frequency and amplitude of Sr$^{2+}$-induced, optogenetically-evoked asynchronous glutamate release from cortical terminals. The frequency of optogenetically-evoked cortico-BLA asynchronous EPSCs (Sr$^{2+}$-EPSCs) decreased significantly, but the amplitude of cortico-BLA Sr$^{2+}$-EPSCs was not altered in slices from PFFs-injected mice (*Figure 4H–J*). Given the unaltered number of vGluT1$^+$ axon terminals and initial release probability of cortico-BLA synapses, fewer readily releasable synaptic vesicles and/or release sites can explain the observed reduction in cortico-BLA transmission.

## αSyn pathology decreases AMPA receptor-mediated current at cortico-BLA synapses

To determine whether the decreased cortico-BLA connection strength was associated with postsynaptic adaptations, we measured the ratio of AMPA- and NMDA-mediated EPSCs (AMPA/NMDA ratio) from the cortico-BLA synapses. In response to optogenetic stimulation, cortico-BLA inputs showed a significant reduction of AMPA/NMDA ratio in slices from PFFs-injected mice relative to controls (*Figure 4K and L*). In addition, we also detected an enhanced inward rectification of AMPA-EPSCs at cortico-BLA synapses in slices from PFFs-injected mice relative to controls (*Figure 4M*), indicating a relatively increased contribution of GluA2-lacking, Ca$^{2+}$-permeable AMPA receptors to cortico-BLA transmission in PFFs-injected mice. These data suggest that αSyn pathology also triggers postsynaptic

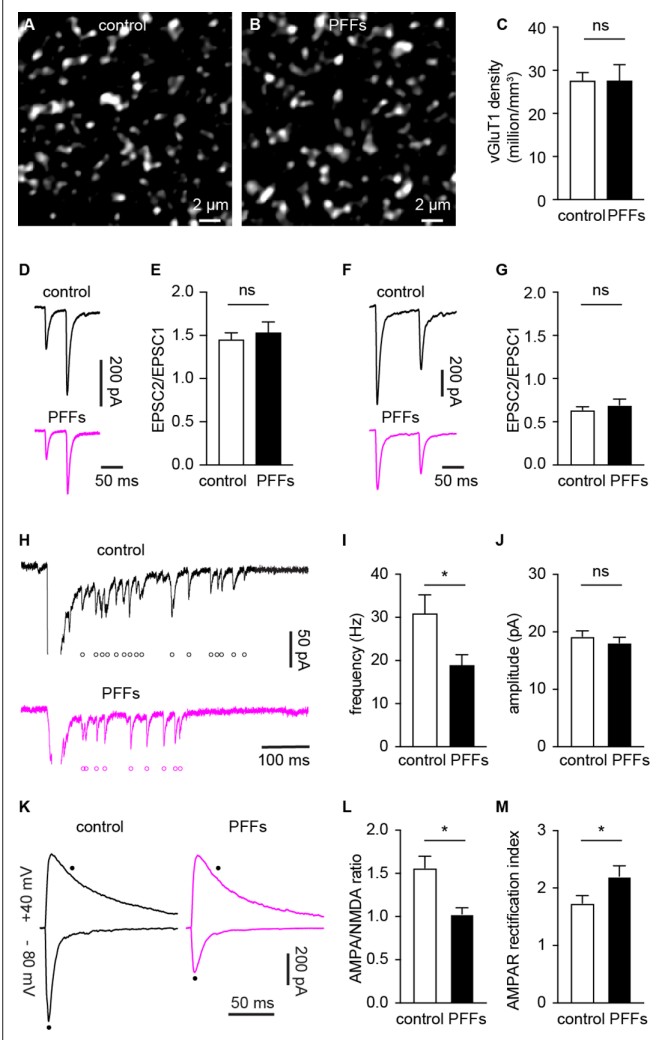

**Figure 4.** α-synuclein (αSyn) pathology decreases the number of functional cortico-BLA inputs. (**A–B**) Representative confocal images showing vesicular glutamate transporter 1 (vGluT1)-immunoreactivity in the basolateral amygdala (BLA) from control (**A**) and PFFs-injected (**B**) mice. (**C**) Summarized data showing no change in the vGluT1 density in the BLA between control and PFFs-injected mice (control=27.5 ± 2 million/mm$^3$, n=11 slices/3 mice; preformed fibrils (PFFs)=27.5 ± 3.8 million/mm$^3$, n=12 slices/3 mice; p=0.98, MWU). (**D–E**) Representative traces of cortico-BLA excitatory postsynaptic currents (EPSCs) evoked by 20 Hz paired pulses of electric stimulation (**D**) and the summarized results of EPSC2/EPSC1 ratios (E, controls=1.45 ± 0.08, n=19 neurons/4 mice; PFFs=1.53 ± 0.13, n=18 neurons/4 mice; p=0.99, MWU). (**F–G**) Representative traces of cortico-BLA EPSCs evoked by 10 Hz paired pulses of optogenetic stimulation (**F**) and the summarized results of EPSC2/EPSC1 ratios (controls=0.64 ± 0.04, n=35 neurons/4 mice; PFFs=0.69 ± 0.07, n=31 neurons/4 mice; p=0.83, MWU). (**H**) Representative traces showing Sr$^{2+}$ induced, optogenetically evoked EPSCs (Sr$^{2+}$-EPSCs) at cortico-BLA synapses from control and PFFs-injected mice. Each open circle indicates a single identified Sr$^{2+}$-EPSC. (**I–J**) Summarized result showing a reduction of the frequency (control=30.9 ± 4.3 Hz, n=24 neurons/3 mice; PFFs=18.9 ± 2.4 Hz, n=24 neurons/3 mice; p=0.012, MWU), but not the amplitude (control=19.2 ± 0.99 pA, n=24 neurons/3 mice; PFF=18.1 ± 0.98 pA, n=24 neurons/3 mice; p=0.31, MWU), of Sr$^{2+}$-EPSCs at cortico-BLA synapses. (**K**) Representative cortico-BLA EPSC traces recorded at –80 mV and +40 mV from control and PFF-injected mice. Black dots indicate the time at which AMPA- and NMDA-mediated components were measured. (**L**) Summarized results showing a decreased AMPA/NMDA ratio at cortico-BLA synapses from PFFs-injected mice relative to controls (control=1.56 ± 0.13, n=20 neurons/3 mice; PFFs=1.03 ± 0.07, n=18 neurons/3 mice, p=0.0012, MWU). (**M**) Summarized results showing an increased AMPA receptor rectification index at cortico-BLA synapses from PFFs-injected mice relative to controls (control=1.74 ± 0.13, n=20 neurons/3 mice; PFFs=2.21 ± 0.18, n=18 neurons/3 mice, p=0.035, MWU). ns, not significant. * p<0.05.

The online version of this article includes the following source data for figure 4:

*Figure 4 continued on next page*

*Figure 4 continued*

**Source data 1.** Source data for plots in *Figure 4*.

adaptations at cortico-BLA synapses. Surprisingly, we did not detect changes in the AMPA/NMDA ratio (control=1.24 ± 0.11, PFFs=1.5 ± 0.12, n=16 neurons/ 3 mice per group, p=0.15, MWU) or AMPA receptor rectification (control=1.12 ± 0.1, PFFs=1.28 ± 0.08, n=16 neurons/ 3 mice per group, p=0.25, MWU) from thalamo-BLA synapse between groups. Thus, the above data suggest that αSyn pathology induces input-specific decrease of postsynaptic AMPA receptor-mediated cortico-BLA transmission, instead of a global reduction of AMPA receptor function in BLA neurons.

Taken together, the development of αSyn pathology selectively decreases the functional cortico-BLA connectivity by inducing both pre- and post-synaptic adaptations and such functional changes occur prior to overt degeneration of axon terminals.

## Pathological aggregation decreases αSyn levels at axon terminals and impairs short-term synaptic plasticity

Formation of cytoplasmic aggregates is believed to move soluble αSyn away from the presynaptic boutons, affecting its role in regulating synaptic vesicle pools (*Benskey et al., 2016*; *Luk et al., 2009*; *Volpicelli-Daley et al., 2011*). Consistent with the earlier predictions, the intensity of αSyn immunoreactivity (Syn1 antibody from BD Biosciences) within vGluT1$^+$ puncta decreased dramatically in slices from PFFs-injected mice, leading to an increased proportion of vGluT1$^+$ boutons that lack a detectable level of αSyn (*Figure 5A–D*). To avoid technical issues associated with immunostaining (e.g. antigen mask due to αSyn aggregation), we assessed the αSyn immunoreactivity within vGluT1$^+$ terminals using a different monoclonal antibody against mouse αSyn–Syn9027 (*Peng et al., 2018*). Consistently, both the αSyn immunoreactivity within vGluT1$^+$ terminals and the proportion of vGluT1$^+$ terminals that are αSyn-immunoreactive decreased significantly in slices from PFFs-injected mice relative to controls (*Figure 5E–H*). The percentage of αSyn-immunoreactive vGluT1$^+$ terminals detected with Syn9027 in PFFs-injected mice was higher than those detected with Syn1, perhaps reflecting differences in the exposure of their epitopes (Syn1: aa 91–99; Syn9027: aa 130–140). These data suggest that αSyn pathology reduces the amount of soluble αSyn present at the cortical axon terminals in the BLA, and in that way could affect its physiological function.

αSyn modulates the dynamics of synapse vesicle pools (*Sulzer and Edwards, 2019*), which plays a key role in regulating synaptic plasticity and the computational function of neural circuits (*Abbott and Regehr, 2004*; *Alabi and Tsien, 2012*). Thus, we stimulated cortico-BLA synapses repetitively to assess the impact of the observed reduction of αSyn levels on short-term synaptic plasticity. We detected a progressive depression of cortico-BLA EPSCs in control mice (*Figure 6A1 and A2*), reflecting mainly a progressive depletion of presynaptic vesicles (*Alabi and Tsien, 2012*; *Cabin et al., 2002*). Interestingly, the cortico-BLA EPSCs in the slices from PFFs-injected mice showed a greater depression relative to those from controls (*Figure 6A1 and A2*). These results are in line with the key role of αSyn in sustaining and mobilizing synaptic vesicle pools (*Cabin et al., 2002*; *Sulzer and Edwards, 2019*; *Vargas et al., 2017*), and suggest the significant impact of the loss of αSyn function on synaptic plasticity and circuit computation.

Because both gains of toxic properties and loss of normal αSyn function are induced in the PFFs models, we employed the αSyn KO mice to further assess the impact of a loss of αSyn on the short-term synaptic plasticity of glutamatergic synapses. Consistently, the amplitude of cortico-BLA EPSCs from αSyn KO mice also exhibited greater depression in response to repetitive stimulation relative to littermate WT controls (*Figure 6B1–B2*), indicating an impaired mobilization of synaptic vesicle pools. It is worth noting that the earlier onset and greater magnitude of cortico-BLA EPSC depression in αSyn KO mice versus that in PFFs-injected mice (*Figure 6A, B*). Thus, it is plausible that the different temporal profiles of short-term synaptic dynamics are linked to the difference in the amount of αSyn present at presynaptic terminals between the PFFs model and KO mice. Last, we did not detect the difference in the temporal profiles of thalamo-BLA EPSCs in slices from αSyn KO mice and those from WT controls (*Figure 6—figure supplement 1*), which indicates a negligible impact of αSyn depletion on synaptic vesicle dynamics and is consistent with the lack of αSyn presence at vGluT2$^+$ axon terminals (*Figure 1*). Together, these results suggest that pathological aggregation decreases the levels

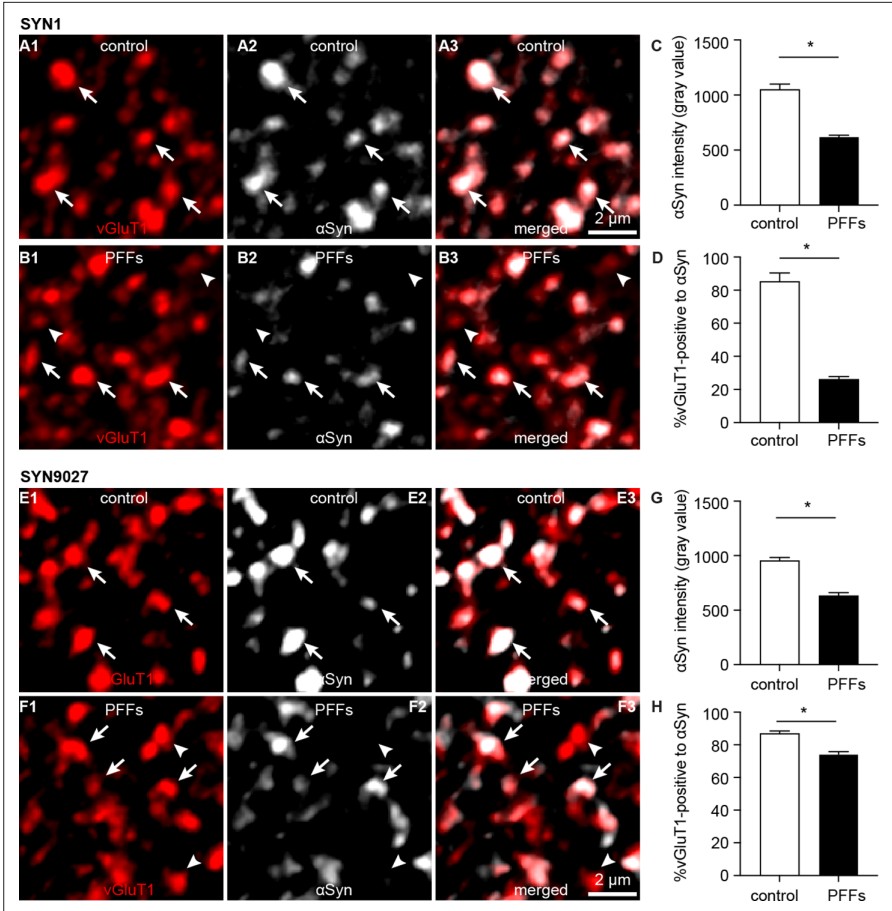

**Figure 5.** Decreased soluble α-synuclein (αSyn) at the axon terminals as pathology develops. (**A1–B3**) Representative confocal images showing αSyn immunoreactivity within vesicular glutamate transporter 1 (vGluT1⁺) puncta in the basolateral amygdala (BLA) using SYN1 antibody from control (**A1–A3**) and PFFs-injected mice (**B1–B3**).(**C**) Summarized graph showing a reduced αSyn immunoreactivity per vGluT1 immunoreactive puncta in PFFs-injected mice relative to controls (control=1049 ± 50; preformed fibrils (PFFs)= 617 ± 18, n=100 puncta/group, p<0.0001, MWU). (**D**) Summarized graph showing a reduced percentage of vGluT1 immunoreactive puncta associated with detectable αSyn immunoreactivity in PFFs-injected mice relative to controls (controls=85 ± 5.2%; PFF=26 ± 1.7%, n=6 slices/group, p=0.002, MWU). (**E1–F3**) Representative confocal images showing αSyn immunoreactivity within vGluT1⁺ puncta in the BLA using SYN9027 antibody from control (**E1–E3**) and PFFs-injected mice (**F1–F3**). (**G**) Summarized graph showing a reduced αSyn immunoreactivity per vGluT1 immunoreactive puncta in PFFs-injected mice relative to controls (control=954 ± 29; PFFs=633 ± 27, n=241 puncta/group, p<0.0001, MWU). (**D**) Summarized graph showing a reduced percentage of vGluT1 immunoreactive puncta associated with detectable αSyn immunoreactivity in PFFs-injected mice relative to controls (controls=87 ± 1.5%, n=12 slices; PFF=74 ± 1.6%, n=11 slices, p<0.0001, MWU).

The online version of this article includes the following source data for figure 5:

**Source data 1.** Source data for plots in **Figure 5**.

of soluble αSyn at the axon terminals, leading to a greater synaptic vesicle depletion and impaired short-term plasticity.

## Discussion

Emerging evidence suggests that αSyn exhibits brain region- and cell type-specific expression in the brain. For example, while αSyn mRNA expression is high in glutamatergic and dopaminergic neurons, it is largely absent from GABAergic neurons across brain regions (**Taguchi et al., 2016**). To further explore such a cell subtype-selective αSyn expression among glutamatergic neurons and synapses, our study shows that αSyn is preferentially present at vGluT1⁺, but not vGluT2⁺, axon terminals across

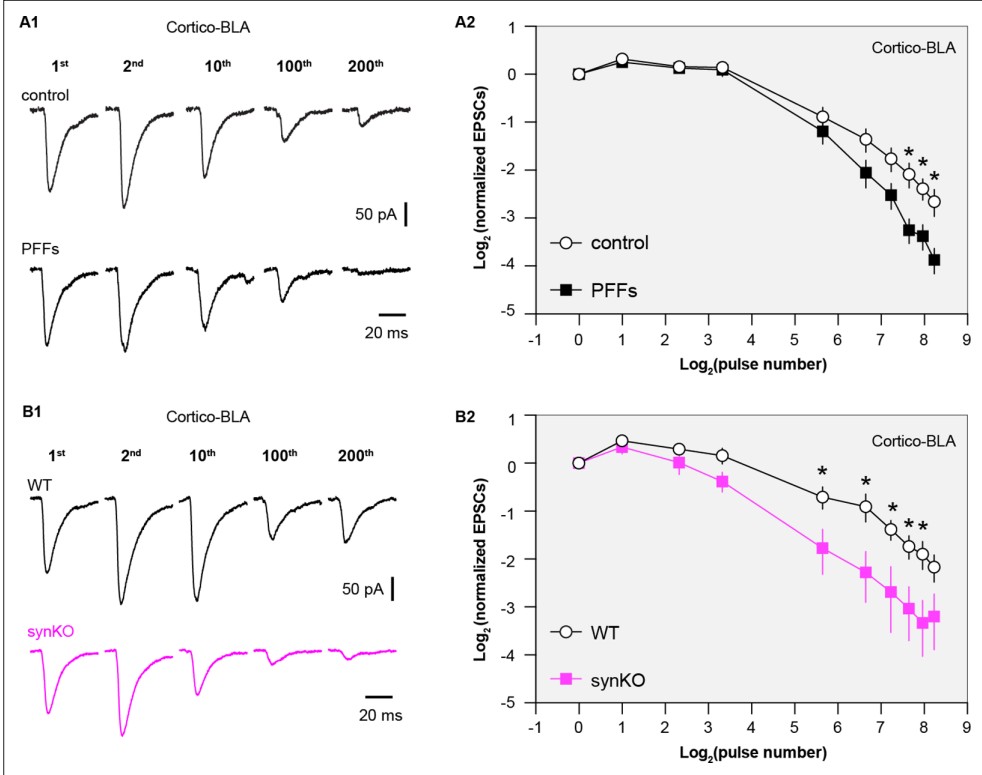

**Figure 6.** Loss of α-synuclein (αSyn) impairs short-term plasticity of the cortico-BLA inputs. (**A1**) Representative cortico-basolateral amygdala (BLA) excitatory postsynaptic currents (EPSCs) traces from control and preformed fibrils (PFF)-injected mice in response to repetitive stimulation (300 stimuli at 12.5 Hz). (**A2**) Summarized graph showing the temporal profiles of cortico-EPSCs depression in slices from control and PFFs-injected mice. The cortico-BLA EPSCs from PFFs-injected mice exhibited greater reduction in the amplitude toward the end of repetitive stimulation. n=13–15 neurons/ 3 mice. (**B1**) Representative cortico-BLA EPSCs traces from wild type (WT) control and αSyn KO mice in response to repetitive stimulation (300 stimuli at 12.5 Hz). (**B2**) Summarized graph showing the temporal profiles of cortico-EPSCs depression in slices from WT control and αSyn KO mice (n=8–9 neurons/3 mice. *, p<0.05, MWU followed by Bonferroni-Dunn correction for multiple comparisons).

The online version of this article includes the following source data and figure supplement(s) for figure 6:

**Source data 1.** Source data for plots in *Figure 6*.

**Figure supplement 1.** α-synuclein (αSyn) KO does not affect short term plasticity profiles of thalamo-BLA inputs.

**Figure supplement 1—source data 1.** Source data for plot in *Figure 6—figure supplement 1*.

several brain regions, including the BLA and the striatum. Consistent with the observation from the axon terminals, we also detected higher amounts of αSyn in the cell bodies of cortical neurons relative to thalamic neurons using the recently generated *Snca*$^{NLS/NLS}$ reporter mice (***Geertsma et al., 2022***). Thus, our results support a cell- and synapse-subtype-selective αSyn protein expression among glutamatergic neurons in the mouse brain.

Endogenous levels of αSyn are determinants of neuronal vulnerability to αSyn pathology. As expected, we observed a decrease in cortical, but not thalamic, inputs to the BLA neurons in a PFF-seeding model of synucleinopathies. At a relatively early-stage (i.e. one-month post-injection), αSyn pathology decreases the number of functional cortical inputs to the BLA without loss of synapse or changes in presynaptic release probability. We posit that the disrupted cortico-BLA transmission can be caused by a reduced size of readily-releasable synaptic vesicle pool and/or a number of presynaptic release sites. This hypothesis was built on a wealth of evidence showing that αSyn aggregates decrease the expression of several synaptic vesicle-associated SNARE proteins (e.g. Snap25 and VAMP2) (***Volpicelli-Daley et al., 2011***). Moreover, our study also suggests a decreased AMPA receptor-mediated response can underlie the impaired cortico-BLA transmission in PFFs-injected mice. On the other hand, an increased relative contribution of GluA2-lacking AMPA receptors can

be a compensatory mechanism for an overall reduction in AMPA-EPSCs at cortico-BLA synapses– a hypothesis that warrants further investigation. It is worth noting that such a decreased AMPA-EPSC of BLA neurons is input specific, which selectively occurs at cortico-BLA synapses. Thus, the molecular mechanisms that cooperate pre- and post-synaptic changes at the cortical inputs, but not the thalamic inputs, remain to be defined.

Our study further highlights the dependence of regional and cellular vulnerability on the endogenous αSyn, i.e., those neurons or axon terminals that express high levels of αSyn are prone to be functionally impacted by αSyn pathology relative to those that lack of or express low levels of αSyn (*Surmeier et al., 2017*; *Thakur et al., 2019*; *Vasili et al., 2022*). Of particular interest is that vGluT2+/ TH+ midbrain dopamine neurons and their axon terminals in the striatum have been shown to be more resilient to neurodegeneration in postmortem PD brains and animal models studies (*Buck et al., 2021*; *Steinkellner et al., 2022*). While several other mechanisms have been proposed (*Buck et al., 2021*), the absence of αSyn at vGluT2+ neurons/terminals could be critical for such resilience.

Loss of normal αSyn function has been thought to be an important but understudied aspect of αSyn pathology. Using an electrophysiological approach, we revealed that a partial or complete removal of αSyn from the axon terminals leads to an enhanced short-term depression in response to repetitive stimulation of cortico-BLA synapses (*Figure 6*). Short-term synaptic plasticity is an important 'gain control' mechanism for neurons to properly balance their responsiveness to distinct afferents in an input-specific manner (*Abbott et al., 1997*). Cortical and thalamic afferents of the BLA exhibit different short-term synaptic plasticity profiles in vivo, which could reflect their different contributions to the formation of emotion-related memory and behavior (*Sigurðsson et al., 2010*). Physiologically, synapse-specific presence of αSyn could underlie such difference in the short-term plasticity profiles of the two inputs (*Figures 1 and 6*).

One can postulate that once aggregates form, a decreased synaptic strength associated with αSyn toxicity (*Figure 3*) and an enhanced synaptic depression due to the loss of αSyn normal function at presynaptic boutons (*Figures 5 and 6*) make the BLA neurons less likely to respond to sustained and repetitive sensory inputs from cortical regions. These circuit changes might explain studies showing a decreased functional connectivity of cortico-amygdala, but not the thalamo-amygdala network in PD patients, which further leads to a failed amygdala responsiveness to the aversive sensory inputs (*Hu et al., 2015*; *Yoshimura et al., 2005*).

# Materials and methods
## Animals
Wild type (WT) C57Bl/6J mice (Jax stock#:000664, RRID: IMSR_JAX:000664) of both sexes (3–4 month-old) were obtained from the Van Andel Institute vivarium internal colony and used in the study. αSyn knockout (*Snca*-/-) mice (RRID: IMSR_JAX:003692) were originally purchased from Jackson laboratories (Bar Harbor, ME) and then were backcrossed on a C57BL/6J background to generate heterozygous *Snca*+/- mice. Experimental *Snca*-/- mice and littermate WT controls were generated from heterozygous *Snca*+/- breeder pairs and were genotyped by Transnetyx (Cordova, TN, USA). *Snca*NLS/ NLS mice (RRID: IMSR_JAX:036763) were generated and maintained on a C57Bl/6J background as described (*Geertsma et al., 2022*). Mice were housed up to four animals per cage under a 12/12 hr light/dark cycle with access to food and water ad libitum in accordance with NIH guidelines for care and use of animals. All animal studies were reviewed and approved by the Institutional Animal Care and Use Committee at Van Andel Institute (animal use protocol#: 22-02-007).

## Preparation and validation of αSyn preformed fibrils
Purification of recombinant mouse αSyn and generation of αSyn preformed fibrils (PFFs) was conducted as described elsewhere (*Luk et al., 2009*; *Volpicelli-Daley et al., 2014*). The pRK172 plasmid (RRID: Addgene_166671) containing the gene of interest was transformed into BL21 (DE3) RIL-competent *E. coli* (230245, Agilent Technologies). A single colony from this transformation was expanded in Terrific Broth (12 g/L of Bacto-tryptone, 24 g/L of yeast extract 4% (v/v) glycerol, 17 mM $KH_2PO_4$, and 72 mM $K_2HPO_4$) with ampicillin. Bacterial pellets from the growth were sonicated and the sample was boiled to precipitate undesired proteins. The supernatant was dialyzed with 10 mM Tris, pH 7.6, 50 mM NaCl, 1 mM EDTA, 1 mM phenylmethylsulfonyl fluoride (PMSF) overnight. Protein was filtered with

a 0.22 μm filter and concentrated using Vivaspin 15 R 10 K centrifugal filters (VS15RH01, Sartorius). Protein was then loaded onto a Superdex 200 column and 2 mL fractions were collected. Fractions were run on SDS-PAGE and stained with InstaBlue protein stain (50-190-5499, Fisher Scientific) to select fractions that were highly enriched in αSyn. These fractions were combined and dialyzed in 10 mM Tris, pH 7.6, 50 mM NaCl, 1 mM EDTA, 1 mM PMSF overnight. Dialyzed fractions were applied to a MonoQ column (HiTrap Q HP 17115401, Cytiva) and run using a NaCl linear gradient (0.025–1 M). Collected fractions were run on SDS-PAGE and stained with InstaBlue protein stain. Fractions that were highly enriched in αSyn were collected and dialyzed into DPBS (Gibco). Protein was filtered through a 0.22 μm filter and concentrated to 7 mg/mL (αSyn) with Vivaspin 15 R 10 K centrifugal filters. Monomer was aliquoted and frozen at –80 °C. For preparation of αSyn PFFs, αSyn monomer was shaken at 1000 rpm for 7 days. Conversion to PFFs was validated by sedimentation at 100,000 x $g$ for 60 min and by thioflavin S staining.

## Stereotaxic surgery

Mice were placed in a stereotaxic frame (Kopf) under 2% isoflurane anesthesia and were supported by a thermostatic heating pad. Recombinant αSyn fibrils were diluted in phosphate-buffered saline (PBS) to a concentration of 5 mg/ml. Prior to stereotaxic injection, PFFs were pulse sonicated at medium intensity with 30 s on and 30 s off for 10 min using a sonicator (Biorupter Pico). To induce the formation of αSyn aggregation in mouse brain (*Luk et al., 2012*), sonicated PFFs (2.0 μl) were injected bilaterally into the dorsal striatum (anteroposterior,+0.2 mm from bregma; mediolateral, ± 2.0 mm from the midline; dorsoventral, –2.6 mm from the brain surface) using a 10 μl syringe (Hamilton) driven by a motorized microinjector (Stoelting) at a rate of 0.2 μl/min. Given the neuronal heterogeneity of the BLA, retrobeads (Lumafluor Inc) were diluted 10 × into αSyn fibrils or PBS to label the projection neurons in the BLA for physiological studies. The final injection volume of PFFs with beads mixture was adjusted to keep a constant amount of αSyn injected across animals. Control mice received 2.0 μl PBS or PBS with retrobeads injections in the same location. For optogenetics studies, AAV vectors encoding ChR2(H134R)-eYFP (titer=1 × 10$^{12}$ vg/ml, Addgene 26973, RRID: Addgene_127090) were stereotaxically injected into and centered at the TeA (anterioposterior –3.3 mm from the bregma, mediodorsal ± 4.1 mm, dorsoventral –1.5 mm from the brain surface) and the midline thalamus (anterioposterior –1.5 mm from the bregma, mediodorsal 0 mm, dorsoventral –3.3 mm from the brain surface) (*Ahmed et al., 2021*; *Amir et al., 2019*). Animals were housed in their home cages before being euthanized for immunohistochemical or physiological studies at one-month post-injection. Details of this protocol can be found at: https://doi.org/10.17504/protocols.io.rm7vzye28l×1/v1.

## Tissue collection – perfusion and sectioning

WT and *Snca*$^{-/-}$ mice received overdosage of avertin intraperitoneally (i.p.) and were subsequently subjected to transcardial perfusion with PBS and 4% paraformaldehyde (PFA, pH 7.4). Brains were removed and post-fixed in 4% PFA overnight, before being re-sectioned at 70 μm using a vibratome (VT1000s, Leica Biosystems, Buffalo Grove, IL). *Snca*$^{NLS/NLS}$ mice were anesthetized with 30 μl of 120 mg/kg Euthanyl (DIN00141704) before being perfused with 10 ml 1 x PBS and 10 ml 4% PFA. Brain tissue was collected and stored for 48 hr in 4% PFA. Brain tissue was then dehydrated in 10, 20, and 30% sucrose solutions for 48 hr each before being flash frozen in –40 °C isopentane for 1 min. Tissues were then cryosectioned at 20 μm and –21°C on the Thermo Scientific HM 525 NX cryostat at the Louise Pelletier Histology Core at the University of Ottawa and stored free floating in 1 × PBS +0.02% NaN3 at 4°C until use. Details of this protocol can be found here.

## Immunofluorescent staining

Brain sections from WT and *Snca*$^{-/-}$ mice were rinsed using PBS and treated with 0.5% Triton X-100% and 2% normal donkey serum (MilliporeSigma) in PBS for 60 min at room temperature, followed by incubation with primary antibodies overnight at room temperature or for 48 hr at 4°C. The concentrations of primary antibodies were rabbit anti-pS129 αSyn (1:10,000, Abcam Cat# 1536–1, RRID: AB_562180), mouse anti-αSyn (1:1000, BD Biosciences Cat# 610787, RRID: AB_398108), mouse anti-NeuN (1:2000, Millipore Cat# MAB377, RRID: AB_2298772), rabbit anti-vGlut1(1:1000, cat#: Millipore Cat# ABN1647, RRID: AB_2814811) and guinea pig anti-vGluT2 (1:1000, Synaptic Systems Cat# 135404, RRID: AB_887884). After being thoroughly rinsed with PBS for 3 times, the sections were

incubated with the secondary antibodies (1:500, AlexaFluor 594 donkey anti-mouse IgG, Jackson ImmunoResearch Labs Cat# 715-586-150, RRID: AB_2340857; AlexaFluor 647 donkey anti-mouse IgG, Jackson ImmunoResearch Labs Cat# 715-607-003, RRID: AB_2340867; AlexaFluor 488 donkey anti-mouse IgG, Jackson ImmunoResearch Labs Cat# 711-546-152, RRID: AB_2340619; AlexaFluor 594 donkey anti-rabbit IgG, Jackson ImmunoResearch Labs Cat# 711-585-152, RRID: AB_2340621, or AlexaFluor 488 donkey anti-guinea pig IgG, Jackson ImmunoResearch Labs Cat# 706-545-148, RRID: AB_2340472) for 90 min at room temperature. Brain sections were rinsed 3 times with PBS and mounted on glass slides using Vectorshield antifade mounting medium (H-1000, Vector Laboratories) and cover slipped.

Brain sections from $Snca^{NLS/NLS}$ were incubated for 24 hr in blocking buffer (1.5% Triton X-100, 10% cosmic calf serum in 1 × PBS), 24 hr in primary Syn1 antibody (1:1000, BD Biosciences Cat# 610787, RRID: AB_398108) and 1 hr in secondary antibody (1:500, Alexa Fluor 568 donkey anti-mouse antibody, (Thermo Fisher Scientific Cat# A10037, RRID: AB_2534013, Lot#: 1917938)) with DAPI at 1:1000 (Millipore Sigma, D9542-1MG). Tissue was washed in 1 × PBS 5 times for 5 min each between each treatment and mounted on Fisherbrand Superfrost Plus slides. After drying for 24 hr, sections were covered with DAKO mounting medium (Cat#: S3023, Lot#: 11347938) and #1.5 coverslips.

Details of immunofluorescent staining can be found here.

## Confocal imaging and analysis

Confocal images from WT and $SNCA^{-/-}$ were acquired using a Nikon A1R confocal microscopy. pS129 αSyn aggregates in different brain regions were imaged under a 40 × objective lens. For αSyn and vGluT1/vGluT2 colocalization analysis, three synaptic markers were immunostained simultaneously and z-stack images were acquired using an oil immersion 100 × objective (NA=1.45; x/y, 1024 × 1024 pixels; z step=150 nm). Images were acquired using identical settings between treatment groups, including laser power, pinhole size, gain, and pixel size. Intensity-based colocalization analysis was performed using Imaris software (RRID: SCR_007370, version 9.3, Oxford, UK, http://www.bitplane.com/imaris/imaris). Background subtraction on z-stack images was conducted using ImageJ (RRID: SCR_003070, NIH, https://imagej.net/) prior to importing files into Imaris. Once imported, two arbitrary regions of interest were created using the surface function (drawing mode: circle; radius=15 μm; number of vertices=30) and the three channels (vGluT2, vGluT1, and αSyn) were masked based on the surface reconstruction to isolate fluorescence within the ROIs. The Imaris 'Coloc' function was used to measure the colocalization between vGluT2/αSyn and vGluT1/αSyn for each ROI. Briefly, either vGluT1 or vGluT2 was selected as channel A and αSyn was selected as channel B. The automatic threshold feature within 'Coloc' was used to calculate the threshold values for each channel. Colocalization channels for vGluT1/ αSyn and vGluT2/αSyn for each ROI were created by using the 'Build Coloc Channel' function. Colocalization parameters were obtained for quantification from the colocalization channels. For quantification of the αSyn intensity within vGluT1$^+$ axon terminals, background subtraction on z-stack images was conducted using ImageJ (RRID: SCR_003070, NIH, https://imagej.net/). After background subtraction, vGluT1 immunoreactive puncta were manually identified as a set of regions of interest (ROI, n=20 puncta per image) from each BLA section and the mean gray values of αSyn immunoreactivity within the same ROI were then measured using ImageJ (RRID: SCR_003070, NIH, https://imagej.net/). Details of images analysis can be found at: dx.doi.org/10.17504/protocols.io.n2bvj61bblk5/v1.

Confocal images from $SNCA^{NLS/SNLS}$ mice were taken on the Zeiss AxioObserver Z1 LSM800 at the Cell Biology and Image Acquisition core at the University of Ottawa. Images were taken at 20 × (0.8 NA) objective with 8bit 1024 × 1024 resolution. The following multichannel acquisition was used to detect signal: DAPI 405 nm/561 nm (650 V); AF568 (αSyn) 405 nm/561 nm (750 V). Images were analyzed using FIJI (RRID: SCR_002285, http://fiji.sc), where Z projected images were separated by channel and the brightness was altered (DAPI: 0–200, αSyn: 0–175). Images were then merged and exported as .jpg files.

To determine cell counts and signal intensity per region, images were inputted into Fiji as 'colorized' images, then Z projected at average intensity. Images were exported as .tiff files and imported into CellProfiler Analyst 3.0 (RRID: SCR_007358, http://cellprofiler.org) (*Stirling et al., 2021*). Image metadata was extracted, and images with 'C' matching 0 were assigned as DAPI, 1 assigned as αSyn. DAPI and αSyn positive cells were counted individually using the 'IdentifyPrimaryObjects' feature

(pixel size 20–50), and αSyn intensity was measured as 'αSyn' from 'DAPI' primary objected. Count and intensity data was analyzed in Prism 9 (RRID: SCR_002798, GraphPad Software, http://www. graphpad.com/). Details of this protocol can be found here.

## Slice preparation for physiology

For physiological studies, mice were deeply anesthetized with avertin (300 mg/kg, i.p.) and then were perfused transcardially with ice-cold, sucrose-based artificial cerebrospinal fluid (aCSF) containing (in mM) 230 sucrose, 26 $NaHCO_3$, 10 glucose, 10 $MgSO_4$, 2.5 KCl, 1.25 $NaH_2PO_4$, and 0.5 $CaCl_2$, 1 sodium pyruvate, and 0.005 L-glutathione. Next, coronal brain slices (300 µm) containing BLA were prepared in the same sucrose-based aCSF solution using a vibratome (VT1200S; Leica Microsystems, Buffalo Grove, IL). Brain slices were kept in normal aCSF (in mM, 126 NaCl, 26 $NaHCO_3$, 10 glucose, 2.5 KCl, 2 $CaCl_2$, 2 $MgSO_4$, 1.25 $NaH_2PO_4$, 1 sodium pyruvate, and 0.005 L-glutathione) equilibrated with 95% $O_2$ and 5% $CO_2$ for 30 min at 35°C and then held at room temperature until use.

## Ex vivo electrophysiology recording and optogenetics

Brain slices were transferred into a recording chamber perfused at a rate of 3–4 ml/min with synthetic interstitial fluid (in mM, 126 NaCl, 26 $NaHCO_3$, 10 glucose, 3 KCl, 1.6 $CaCl_2$, 1.5 $MgSO_4$, and 1.25 $NaH_2PO_4$) equilibrated with 95% $O_2$ and 5% $CO_2$ at 33–34°C via a feedback-controlled in-line heater (TC-324C, Warner Instruments) (*Chen et al., 2021*). SR-95531 (GABAzine, 10 µM) was routinely added extracellularly to block $GABA_A$ receptor-mediated inhibitory synaptic transmission. Neurons were visualized and recorded under gradient contrast SliceScope 1000 (Scientifica, Uckfield, UK) with infrared illumination using an IR-2000 CCD camera (DAGE-MTI, USA) and motorized micromanipulators (Scientifica, Uckfield, UK). Individual BLA projection neurons labeled with retrobeads were identified using a 60 × water immersion objective lens (Olympus, Japan) and targeted for whole-cell patch-clamp recording. Data were collected using a MultiClamp 700B amplifier and a Digidata 1550B digitizer at a sampling rate of 50 kHz under the control of pClamp11 (RRID: SCR_011323, Molecular Devices, San Jose, USA). Borosilicate glass pipettes (O.D.=1.5 mm, I.D.=0.86 mm, item #BF150-86-10, Sutter Instruments) for patch clamp recordings (4–6 MΩ) were pulled using a micropipette puller (P1000, Sutter Instruments, Novato, CA).

To assess glutamatergic transmission in the BLA, glass pipettes were filled with a cesium methanesulfonate based internal solution of (in mM): 120 $CH_3O_3SCs$, 2.8 NaCl, 10 HEPES, 0.4 $Na_4$-EGTA, 5 QX314-HBr, 5 phosphocreatine, 0.1 spermine, 4 ATP-Mg, and 0.4 GTP-Na (pH 7.3, 290 mOsm). BLA neurons were voltage clamped at –70 mV to assess the EPSCs in response to presynaptic electrical or optogenetic stimulations. Concentric bipolar electrodes (FHC, Bowdoin, ME) or glass pipettes (~2 MΩ) filled with the extracellular solution were used as stimulating electrodes and placed on the external and internal capsules to evoke glutamate release in the BLA from cortical and thalamic axon terminals, respectively (*Shin et al., 2010*). A range of electrical pulses (intensities=30–120 µA, duration=100 µs) were delivered through a stimulator (Digitimer, UK) to evoke glutamate release at either cortical or thalamic inputs. To study the process of synaptic vesicle pool mobilization during repetitive glutamatergic transmission, we delivered 300 electrical pulses (duration=100 µs) at 12.5 Hz and measured the amplitudes of EPSCs to quantify changes in the presynaptic glutamate release (*Cabin et al., 2002*). Only one neuron was recorded from each slice in the prolonged repetitive stimulation studies.

Optogenetic stimulation pulses (1 ms duration) were delivered through a 60 × objective lens (Olympus, Japan) using a 470 nm LED light source (CoolLED, UK). To isolate monosynaptic cortico-BLA and thalamo-BLA EPSCs, optogenetically evoked EPSCs were recorded in the presence of TTX (1 µM) and 4-AP (100 µM). Series resistance (Rs <20 MΩ) was regularly monitored throughout the recording to ensure Rs changes were less than 15%. Liquid junction potential (~9 mV) was not corrected. Details of the protocol can be found here.

## Data analysis and statistics

Electrophysiology data were analyzed offline in Clampfit 11.1 (RRID: SCR_011323, Molecular Devices). The peak amplitude of monosynaptic EPSCs in response to electric stimulation was quantified from an average of three to five sweeps. Digital confocal images were analyzed using ImageJ (RRID: SCR_003070, NIH, https://imagej.net/) or Imaris (RRID: SCR_007370, Oxford, UK, http://www.bitplane.

com/imaris/imaris). Statistics were performed using Prism9 (RRID: SCR_002798, GraphPad Software, http://www.graphpad.com/). Non-parametric, distribution-independent Mann–Whiney U (MWU) test was used for non-paired data comparisons between groups, followed by Bonferroni-Dunn correction for multiple comparisons. Data from $Snca^{SNL/SNL}$ mice were compared using one-way AVOVA (Šídák's multiple comparisons test). All tests were two-tailed, and <i>p-value<0.05 was considered statistically significant. Summary results are reported as mean plus standard error of mean.

## Acknowledgements

This work was supported by the NARSAD Young Investigator award from the Brain and Behavior Research Foundation (HYC) and an investigator-initiated research award from the Department of Defense (W81XWH2110943, PI: HYC). Dr. Hong-Yuan Chu is a Frederick & Alice Coles and Thomas & Nancy Coles Investigator. This research was funded in part by Aligning Science Across Parkinson's [ASAP-020625 for MWCR, ASAP-020616 for MXH, ASAP-020572 for HYC] through the Michael J Fox Foundation for Parkinson's Research (MJFF). For the purpose of open access, the author has applied a CC BY public copyright license to all Author Accepted Manuscripts arising from this submission. The authors thank the Van Andel Institute Optical Imaging Core for the advanced confocal microscopy as well as the Van Andel Institute Vivarium and Transgenics Core, especially Megan Tompkins, Alyssa Bradfield, Malista Powers, and William Weaver for animal husbandry and colony maintenance.

## Additional information

### Competing interests

Patrik Brundin: has received support as a consultant from AbbVie, Axial Therapeutics, Calico Life Sciences, CuraSen, Enterin Inc, Fujifilm-Cellular Dynamics International, Idorsia Pharmaceuticals, Lundbeck A/S. He has received commercial support for research from Lundbeck A/S and F. Hoffman-La Roche. He has ownership interests in Acousort AB, Axial Therapeutics, Enterin Inc and RYNE Biotechnology. During the time that this paper was undergoing revision he became an employee of F. Hoffman-La Roche, although none of the data were generated by this company. The other authors declare that no competing interests exist.

### Funding

| Funder | Grant reference number | Author |
|---|---|---|
| Brain and Behavior Research Foundation | | Hong-Yuan Chu |
| Congressionally Directed Medical Research Programs | | Hong-Yuan Chu |
| Aligning Science Across Parkinson's | ASAP-020616 | Michael X Henderson Maxime WC Rousseaux Hong-Yuan Chu |
| Aligning Science Across Parkinson's | ASAP-020625 | Maxime WC Rousseaux |
| Aligning Science Across Parkinson's | ASAP-020572 | Hong-Yuan Chu |

The funders had no role in study design, data collection and interpretation, or the decision to submit the work for publication.

### Author contributions

Liqiang Chen, Chetan Nagaraja, Samuel Daniels, Data curation, Formal analysis, Investigation, Validation; Zoe A Fisk, Formal analysis, Investigation, Visualization; Rachel Dvorak, Formal analysis; Lindsay Meyerdirk, Jennifer A Steiner, Resources; Martha L Escobar Galvis, Patrik Brundin, Conceptualization, Visualization, Writing – review and editing; Michael X Henderson, Resources, Writing – review and editing; Maxime WC Rousseaux, Methodology, Resources, Writing – review and editing; Hong-Yuan

Chu, Conceptualization, Data curation, Formal analysis, Funding acquisition, Investigation, Methodology, Project administration, Resources, Supervision, Validation, Visualization, Writing – original draft, Writing – review and editing

### Author ORCIDs
Liqiang Chen http://orcid.org/0000-0003-3236-1129
Jennifer A Steiner http://orcid.org/0000-0003-0953-1310
Martha L Escobar Galvis http://orcid.org/0000-0001-8400-9392
Hong-Yuan Chu http://orcid.org/0000-0003-0923-683X

### Ethics
All animal studies were reviewed and approved by the Institutional Animal Care and Use Committee at Van Andel Institute (animal use protocol#: 22-02-007).

### Decision letter and Author response
Decision letter https://doi.org/10.7554/eLife.78055.sa1
Author response https://doi.org/10.7554/eLife.78055.sa2

---

## Additional files

### Supplementary files
• Transparent reporting form

### Data availability
All source data associated with the revised manuscript have been deposited on Open Science Framework: https://doi.org/10.17605/OSF.IO/264SM. All data generated or analyzed during this study are included in the manuscript and source data have been provided for all main and supplementary figures.

The following dataset was generated:

| Author(s) | Year | Dataset title | Dataset URL | Database and Identifier |
| --- | --- | --- | --- | --- |
| Chu H-Y | 2022 | Synaptic Location Is a Determinant of the Detrimental Effects of α-Synuclein Pathology to Glutamatergic Transmission in the Basolateral Amygdala | https://doi.org/10.17605/OSF.IO/264SM | Open Science Framework, 10.17605/OSF.IO/264SM |

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
