## [Editor Report]

The manuscript by Chen et al., examines the synapse-specificity of α-synuclein aggregation and corresponding circuit dysfunction in the amygdala. Using confocal microscopy and slice electrophysiology, along with α-synuclein knockout mice and preformed fibrils, the authors demonstrate that cortico-amygdala, but not thalamo-amygdala, inputs are more vulnerable to α-synuclein aggregation and corresponding synaptic dysfunction. This has important implications for the etiology of psychiatric deficits that are common in Parkinson's disease.

---

## [Decision Letter]

**Decision letter after peer review:**

Thank you for submitting your article "Synaptic Location Is a Determinant of the Detrimental Effects of α-Synuclein Pathology to Glutamatergic Transmission in the Basolateral Amygdala" for consideration by *eLife*. Your article has been reviewed by 3 peer reviewers, one of whom is a member of our Board of Reviewing Editors, and the evaluation has been overseen by a Reviewing Editor and Lu Chen as the Senior Editor. The reviewers have opted to remain anonymous.

Essential revisions:

1. The authors need to provide additional evidence of pathologic α-synuclein in the amygdala- preferably at synaptic terminals.

2. Additional controls are needed to support the conclusion – a reduction in synaptic α-synuclein. Because the antigen could be masked by aggregation, it is important to verify the same results using an antibody to a different α-synuclein epitope.

3. Please provide evidence on whether the defect could be rescued with a cortically injected/expressed α-syn AAV.

4. Additional experiment – for example, looking at paired-pulse ratios of EPSCs – is needed to further support the conclusion on presynaptic release deficits.

*Reviewer #1 (Recommendations for the authors):*

The selective localization of α-synuclein to vGLUT1, but not vGLUT2, positive terminals is convincing. The effect of PFFs on EPSCs in the amygdala and on short-term depression in cortical-amygdala synapses is also convincing. This is really the first study to my knowledge that really shows that pathologic α-synuclein disrupts amygdala physiology and is therefore very important for the field.

However, the authors try to claim that the defects are caused by the depletion of normal α-synuclein in vGLUT1 terminals. The main technique used for this was immunofluorescence with one synuclein antibody. It is possible that the formation of aggregates masks the epitope for the antibody used. Does IF with another synuclein antibody show the same results? Does increasing expression of synuclein at these terminals rescue the defects?

There is no data that there are abnormal aggregates in the amygdala. Can the authors please show proteinase K digested tissue with an antibody to total synuclein or use an antibody to phosphorylated synuclein?

Also, the mechanism of how potential abnormal α-synuclein causes reduced short-term plasticity is unclear. Did the aggregates sequester vesicles and prevent release from the reserve pool? Was the RRP diminished?

Excellent work has shown that normal α-synuclein acts as a clamp-on vesicle release- how does this data support or refute these studies (PMID: 31110017)?

The use of the PFF model and α-synuclein KO mice was well-chosen however, PBS as a control for a protein injection model might not be the best fit. Consider a monomer injected mouse as another negative control to control for injections of protein.

*Reviewer #2 (Recommendations for the authors):*

1. Page 1 (starting with Title page) line 144: "buttons" should be boutons.

2. The Discussion section would probably benefit from a little expansion. Along these lines, there are some points in the results that may be better left for the discussion (for example, much of the second half of the second paragraph on page 10).

3. Page 11 line 175- clarify if it is the amount of "soluble" α-synuclein that's reduced in cortical axon terminals.

4. Page 13 line 203: "temporary" should be temporal.

5. Page 13 line 207: "neglectable" should be negligible.

*Reviewer #3 (Recommendations for the authors):*

1. The authors stated that "One-month post-injection, we detected robust 122 αSyn pathology in vGluT1^+^ cerebral cortical regions (e.g., the temporal association cortex (TeA) and the perirhinal cortex), but we barely observed any cytoplasmic aggregates in vGluT2^+^ thalamic regions (data not shown), which is consistent with earlier reports".

"did not detect difference in the cortico-BLA transmission between PFFs152 versus PBS-injected αSyn KO mice (data not shown).

Data not shown – This should be replaced by real data.

2. What are the endogenous expression levels of a-syn in the thalamus? Please provide a quantitative description.

3. Because the striatum also receives convergent cortical and thalamic inputs, it would strengthen the conclusion if the authors systematically investigate corticostriatal vs thalmostriatal terminals in parallel.

4. The authors used optogenetic tools to verify the results using electrical stimulation of external and internal capsules. It would be helpful if release probability is also compared using optogenetic stimulation. Indeed, the desensitization properties of ChR2 may complicate the readout. However, the relative change in EPSC amplitudes would further strengthen the claim.

---

## [Author Response]

Essential revisions:1. The authors need to provide additional evidence of pathologic α-synuclein in the amygdala- preferably at synaptic terminals.

Immunoreactivity of pathologic pS129 αSyn in the BLA has now been included in the Figure 3B, in which we show both Lewy body- and Lewy neurite-like aggregates in the BLA. Please note the distribution of pS129 αSyn immunoreactivity within the external capsule, where cortical afferents enter BLA (Figure 3B, C), indicating the formation and accumulation of αSyn pathology along the axons arising from the cortical regions. We also performed co-staining of vGluT1 and pS129 αSyn. Their immunofluorescent signals did not show co-localization, suggesting that aggregation perhaps concentrated at soma and axons. This observation is consistent with evidence in the literature (Volpicelli-Daley et al., 2011), showing pS129 pathology was more prominent within axons of cultured neurons.

2. Additional controls are needed to support the conclusion – a reduction in synaptic α-synuclein. Because the antigen could be masked by aggregation, it is important to verify the same results using an antibody to a different α-synuclein epitope.

To further confirm the reduction of soluble αSyn at synaptic terminals, we conducted additional immunostaining using syn9027, a monoclonal antibody against mouse α-synuclein (Peng et al., 2018). Consistent with our initial observations, reduction of both αSyn immunofluorescence within vGuT1^+^ axon terminals and the proportion of vGluT1 showing αSyn immunoreactivity were also detected using syn9027 antibody (Figure 5E-H). The percentage of αSyn-immunoreactive vGluT1^+^ terminals detected with Syn9027 in PFFs-injected mice was higher than those detected with Syn1, perhaps reflecting differences in the exposure of their epitopes (Syn1: aa 91-99; Syn9027: aa 130-140). We thank the reviewer again for raising this important question.

3. Please provide evidence on whether the defect could be rescued with a cortically injected/expressed α-syn AAV.

The authors thank the reviewers and editors for raising this question. We agree that successful execution of such an experiment could provide a definitive answer to the question that rises from our work – are the changes in synaptic plasticity related to the decrease in available αSyn at the synapse? Unfortunately, there are technical challenges that prevent us of conducting such experiment.

– First, there is a large body of in vitro evidence suggesting that αSyn concentration needs to be properly maintained at synaptic terminals for its physiological function (i.e., “synuclein homeostasis”) (Dettmer et al., 2016). Also, the propensity of soluble αSyn to aggregate in the presence of PFFs has been shown to be concentration dependent (Courte et al., 2020; Vasili et al., 2022), which would be an inherent problem if we were to increase levels of soluble αSyn after PFFs injections have triggered the formation of aggregates in the brain. The same applies to the AAV approach, which is known to increase αSyn levels in rodents eventually resulting in its aggregation; AAV-mediated expression of αSyn in conjunction with PFFs inoculation augments the development of αSyn pathology (Thakur et al., 2017). The combined effect of AAV-mediated overexpression and recruitment of αSyn into aggregates would make it hard to determine the level of soluble αSyn that could be maintained at the synaptic terminals.

– Second, the BLA receives glutamatergic inputs from broad cortical regions. The present work electrically stimulated the external capsules to activate these cortical inputs and revealed deficits in the short-term plasticity of cortico-BLA synapses of PFFs injected mice (Figure 6). To rescue such deficits in synaptic plasticity using repetitive electric stimulation (*optogenetics would not work due to ChR2 inactivation*), synaptic levels of αSyn should be brought to “normal levels” across wide cortical regions. To achieve this goal, multiple injections to cover most of the cerebral cortex will be required, and the variability associated with such multi-site manipulation would be unpredictable, not only because of the possible differences in the amounts of injected material but also because αSyn levels could be affected by the processes triggered by the injection *per se* (i.e., inflammatory response(Acosta et al., 2015)).

– Last, the “physiological” level of soluble αSyn at axon terminals remains undefined and unmeasurable in vivo, so even in the case that we were to succeed in increasing the levels of αSyn at the synapse, failure to rescue the functional deficits would not be a definitive answer. The increase could be simply not sufficient to rescue the deficits; this situation will be analogous to the PFFs injected animals when compared to the αSyn KO mice.

Altogether, in vivo rescue studies will require not only extensive optimization of the experimental conditions but also access to technologies and resources that are not currently available

4. Additional experiment – for example, looking at paired-pulse ratios of EPSCs – is needed to further support the conclusion on presynaptic release deficits.

To support further presynaptic release deficits, we performed additional experiments/analyses (Figure 4), including:

– Quantification of vGluT1^+^ terminals using stereology (Figure 4A-C), which shows no difference in vGluT1 density between groups and suggests no degeneration of cortical axon terminals in the BLA.

– Paired-pulse ratios (PPRs) of cortico-BLA synapses using both electric and optogenetic stimulation (Figure 4D-G). No difference was detected in PPRs between groups, indicating that the decreased cortico-BLA transmission was not due to a lower presynaptic initial release probability.

– Analyses of quantal release: We analyzed Sr^2+^-induced, optogenetically-evoked quantal glutamate release from cortico-BLA synapses (Figure 4H-J) and detected a significant reduction in the frequency, but not the amplitude, of Sr^2+^-induced EPSCs in PFFs-injected mice.

Combining evidence in the literature, we posit that reduced presynaptic release sites and/or number of synaptic vesicles likely underlie the disrupted cortico-BLA transmission associated with αSyn pathology. These new results and discussion can be found on Pages 13-17.

Reviewer #1 (Recommendations for the authors):The selective localization of α-synuclein to vGLUT1, but not vGLUT2, positive terminals is convincing. The effect of PFFs on EPSCs in the amygdala and on short-term depression in cortical-amygdala synapses is also convincing. This is really the first study to my knowledge that really shows that pathologic α-synuclein disrupts amygdala physiology and is therefore very important for the field.

The authors thank the reviewer for the kind words.

However, the authors try to claim that the defects are caused by the depletion of normal α-synuclein in vGLUT1 terminals. The main technique used for this was immunofluorescence with one synuclein antibody. It is possible that the formation of aggregates masks the epitope for the antibody used. Does IF with another synuclein antibody show the same results? Does increasing expression of synuclein at these terminals rescue the defects?

In the revised manuscript, we further clarify that the decreased cortico-BLA basal transmission and the abnormal short-term plasticity are likely due to both the gained toxic properties of αSyn aggregation and the loss of αSyn normal function. Also, we have followed the Reviewer’s suggestion and performed additional immunostaining using a different monoclonal antibody Syn9027 against αSyn (Peng et al., 2018). We confirmed (1) a reduced proportion of vGluT1^+^ terminals that are αSyn immunoreactive; and (2) a decreased αSyn immunoreactivity at vGluT1^+^ axon terminals in slices from PFFs-injected mice relative to those from controls (Figure 5E-H). Thus, immunostaining studies using two different antibodies support a reduced soluble αSyn at cortical axon terminals. Please also see responses to the essential revision #2 above.

In addition, cortico-BLA synapses from slices of PFFs-injected mice showed enhanced short-term depression, which recapitulated key kinetic feature of a complete αSyn depletion in SynKO mice (Figure 6). Combining immunohistochemical and physiological information, we posit that deficits in short-term plasticity of cortical-BLA synapses in PFFs-injected mice are largely caused by the depletion of normal αSyn from axon terminals. However, as addressed in essential revision #3 (see above) it is technically impossible to increase αSyn levels without causing toxic effects in the PFFs mouse model.

There is no data that there are abnormal aggregates in the amygdala. Can the authors please show proteinase K digested tissue with an antibody to total synuclein or use an antibody to phosphorylated synuclein?

We agree with the reviewer that pathologic αSyn aggregates should have been documented in the initial submission. We have included experimental data showing the accumulation of pathologic Ser129-phosphorylated αSyn in the BLA– both Lewy body- and Lewy neurite-like structures were visualized (Figure 3). Please also see the response to the essential revision #1.

Also, the mechanism of how potential abnormal α-synuclein causes reduced short-term plasticity is unclear. Did the aggregates sequester vesicles and prevent release from the reserve pool? Was the RRP diminished?

In the present study, we used the prolonged electric repetitive stimulation protocol (300 pulses@12.5Hz) to gradually discharge readily releasable pool (RRP, the early phase) and recruit the reserve pool (the late phase) of synaptic vesicles (SVs) into the cycle (Alabi and Tsien, 2012; Cabin et al., 2002). We did not detect statistically significant differences in the temporal profiles of cortico-BLA EPSCs to the first 100 stimuli between PBS- and PFFs-injected mice (Figure 6). This observation indicates the RRP at cortico-BLA synapses perhaps is likely not diminished, which agrees with the data from morphology study by Froula et al., (2018) (Froula et al., 2018). Thus, we posited that development of αSyn pathology does not diminish RRP at cortico-BLA synapses. Interestingly, we noticed that the depression of cortico-BLA synapses was enhanced toward the late phase of the repetitive stimulation in slices from PFFs-injected mice compared to controls (Figure 6), indicating an impaired SVs refilling process once RRP is depleted during repetitive stimulation.

Consistent with evidence from hippocampal slices, we also observed stronger depression of cortico-BLA EPSCs in response to repetitive stimulation in slices from αSyn KO mice. Such a progressive suppression of glutamate release in the absence of αSyn has been linked with a depletion of synaptic vesicle pool (Cabin et al., 2002). Moreover, we detected a decreased immunoreactivity of soluble αSyn at the vGluT1^+^ axon terminals in the BLA (Figure 5). Furthermore, PFFs-mediated αSyn aggregation decreases the expression of synapsins (Volpicelli-Daley et al., 2011), and loss of synapsins function is known to decrease SVs numbers (Siksou et al., 2007). Thus, it is plausible that the size of SVs reserve pool is smaller in the PFFs models due to both a loss of normal αSyn function and a consequence of its gained toxic properties. A decreased SVs reserve pool or impaired mobility can slow RRP refilling during repetitive stimulation and lead to an enhanced short-term depression (Alabi and Tsien, 2012).

In conclusion, we propose that the enhanced short-term depression observed in the mice with αSyn pathology was mainly caused by a decreased SVs reserve pool and/or an impaired refilling of RRP at cortico-BLA synapses.

Excellent work has shown that normal α-synuclein acts as a clamp-on vesicle release- how does this data support or refute these studies (PMID: 31110017)?

We agree that the elegant work by Sun *et al.,* demonstrated a functional cooperation of αSyn and VAMP2 (PMID: 31110017). It is worth noting that the work by Sun *et al.,* was built on an in vitro system that employs αSyn overexpression, making the levels of αSyn non-physiological; overexpression also promotes the formation of αSyn oligomers, which eventually leads to pathologic aggregation. While αSyn-VAMP2 binding was detected, whether the αSyn-VAMP2 interaction *per se*, and/or other high order αSyn species, mediates the SVs attenuation remains unclear.

Further, the technical differences between the present work and Sun *et al.*, such as in vivo versus in vitro models, levels of αSyn species in terminals and soma (endogenous or overexpression) and its toxic species, temporal resolution of the used experimental approaches/outcomes (optical imaging versus electrophysiology) make a direct comparation difficult.

The use of the PFF model and α-synuclein KO mice was well-chosen however, PBS as a control for a protein injection model might not be the best fit. Consider a monomer injected mouse as another negative control to control for injections of protein.

We agree with the reviewer that αSyn monomer injection could be an additional negative control for injection of protein in PFFs models. There is a vast body of literature using PFFs models showing no difference in pathology development between monomeric αSyn- and PBS-injected animals, so most studies combined two treatments as a “control” group (e.g., (Chatterjee et al., 2019; Kim et al., 2019; Luk et al., 2009; Rey et al., 2016)). Since our main goal was to focus on the function impact of the pathology, we opted for PBS as control because it represents a “cleaner” option (i.e., monomer preparations can in some cases elicit proinflammatory responses associated with presence of endotoxins). In addition, in the present study, we did not detect αSyn pathology and the subsequent impairments of cortico-BLA transmission when PFFs were injected into syn KO mice (Figure 3—figure supplementary 1), providing further evidence that injection of protein *per se* was not be sufficient to impair basal cortico-BLA transmission. Therefore, adding an additional control group with monomeric αsyn injection would not change the conclusion. The authors sincerely request to use the available resources to work on other key studies but will include monomer-injected controls in future follow-up studies.

Reviewer #2 (Recommendations for the authors):1. Page 1 (starting with Title page) line 144: "buttons" should be boutons.

Thanks for pointing out this error. “buttons” have been corrected to “boutons” throughout the manuscript.

2. The Discussion section would probably benefit from a little expansion. Along these lines, there are some points in the results that may be better left for the discussion (for example, much of the second half of the second paragraph on page 10).

The initial submission was formatted as “short report”, so we combined “results” and “discussion” sections. For this revision, the manuscript has been re-formatted the manuscript into full “research article” and have separated “discussion” from “results” section.

3. Page 11 line 175- clarify if it is the amount of "soluble" α-synuclein that's reduced in cortical axon terminals.

We thank the reviewer for the thoughtful comments. The sentence has been clarified as a reduction of “soluble α-synuclein” in cortical axon terminals.

4. Page 13 line 203: "temporary" should be temporal.

It has been corrected in the revised manuscript.

5. Page 13 line 207: "neglectable" should be negligible.

It has been corrected in the revised manuscript.

Reviewer #3 (Recommendations for the authors):1. The authors stated that "One-month post-injection, we detected robust 122 αSyn pathology in vGluT1^+^ cerebral cortical regions (e.g., the temporal association cortex (TeA) and the perirhinal cortex), but we barely observed any cytoplasmic aggregates in vGluT2^+^ thalamic regions (data not shown), which is consistent with earlier reports"."did not detect difference in the cortico-BLA transmission between PFFs152 versus PBS-injected αSyn KO mice (data not shown).Data not shown – This should be replaced by real data.

The authors thank the reviewer for the thoughtful comments. We have updated the revised manuscript, which is now being submitted as a full “research article” and that shows the level of αSyn pathology in the cortical and thalamic regions (Figure 3A-C). We also have added the data to show PFFs-injection into the striatum of αSyn KO mice did not alter cortico-BLA transmission (Figure 3—figure supplementary 1).

2. What are the endogenous expression levels of a-syn in the thalamus? Please provide a quantitative description.

αSyn immunoreactivity from WT mice gives diffused signals due to its presynaptic enrichment. Such staining would be ideal to tell the endogenous expression levels of αSyn protein in the thalamus. To solve this issue, we used *Snca^NLS/NLS^* reporter mice to determine αSyn protein localization in the cortical and thalamic areas. These mice localize endogenous αSyn to the nucleus, allowing the visualization of cellular topography (Figure 2A) (Geertsma et al., 2022). We detected significantly lower αSyn expression in the thalamus compared to the cortex, although there is subregion heterogeneity in αSyn expression levels in the thalamus. Particularly, the centromedial thalamaus that projects heavily to the BLA shows an absence of αSyn expression; these observations provide further support to of our physiology data (Figure 2B-D).

3. Because the striatum also receives convergent cortical and thalamic inputs, it would strengthen the conclusion if the authors systematically investigate corticostriatal vs thalmostriatal terminals in parallel.

Please see the response to point #4 of this reviewer’s Public Review.

4. The authors used optogenetic tools to verify the results using electrical stimulation of external and internal capsules. It would be helpful if release probability is also compared using optogenetic stimulation. Indeed, the desensitization properties of ChR2 may complicate the readout. However, the relative change in EPSC amplitudes would further strengthen the claim.

In the revised manuscript, we employed both electrical and optogenetic stimulations to assess the potential effects of αSyn pathology on the initial release probability at cortico-BLA synapses. We did not detected changes in PPRs using either approach (Figure 4D-G). These data and related discussion have been included in the revised manuscript. We concluded that PFFs-induced αSyn aggregation does not alter the initial release probability of SVs at cortico-BLA terminals.

References

Acosta SA, Tajiri N, Pena I de la, Bastawrous M, Sanberg PR, Kaneko Y, Borlongan CV. 2015. Α-Synuclein as a Pathological Link Between Chronic Traumatic Brain Injury and Parkinson’s Disease. *J Cell Physiol* 230:1024–1032. doi:10.1002/jcp.24830

Alabi AA, Tsien RW. 2012. Synaptic Vesicle Pools and Dynamics. *Csh Perspect Biol* 4:a013680. doi:10.1101/cshperspect.a013680

Cabin DE, Shimazu K, Murphy D, Cole NB, Gottschalk W, McIlwain KL, Orrison B, Chen A, Ellis CE, Paylor R, Lu B, Nussbaum RL. 2002. Synaptic Vesicle Depletion Correlates with Attenuated Synaptic Responses to Prolonged Repetitive Stimulation in Mice Lacking α-Synuclein. *J Neurosci* 22:8797–8807. doi:10.1523/jneurosci.22-20-08797.2002

Chatterjee D, Sanchez DS, Quansah E, Rey NL, George S, Becker K, Madaj Z, Steiner JA, Ma J, Galvis MLE, Kordower JH, Brundin P. 2019. Loss of One Engrailed1 Allele Enhances Induced α-Synucleinopathy. *J Park Dis* 9:315–326. doi:10.3233/jpd-191590

Courte J, Bousset L, Boxberg YV, Villard C, Melki R, Peyrin J-M. 2020. The expression level of α-synuclein in different neuronal populations is the primary determinant of its prion-like seeding. *Sci Rep-uk* 10:4895. doi:10.1038/s41598-020-61757-x

Dettmer U, Selkoe D, Bartels T. 2016. New insights into cellular α-synuclein homeostasis in health and disease. *Curr Opin Neurobiol* 36:15–22. doi:10.1016/j.conb.2015.07.007

Froula JM, Henderson BW, Gonzalez JC, Vaden JH, Mclean JW, Wu Y, Banumurthy G, Overstreet-Wadiche L, Herskowitz JH, Volpicelli-Daley LA. 2018. α-Synuclein fibril-induced paradoxical structural and functional defects in hippocampal neurons. *Acta Neuropathologica Commun* 6:35. doi:10.1186/s40478-018-0537-x

Geertsma HM, Suk TR, Ricke KM, Horsthuis K, Parmasad J-LA, Fisk ZA, Callaghan SM, Rousseaux MWC. 2022. Constitutive nuclear accumulation of endogenous α-synuclein in mice causes motor impairment and cortical dysfunction, independent of protein aggregation. *Hum Mol Genet*. doi:10.1093/hmg/ddac035

Kim S, Kwon S-H, Kam T-I, Panicker N, Karuppagounder SS, Lee S, Lee JH, Kim WR, Kook M, Foss CA, Shen C, Lee H, Kulkarni S, Pasricha PJ, Lee G, Pomper MG, Dawson VL, Dawson TM, Ko HS. 2019. Transneuronal Propagation of Pathologic α-Synuclein from the Gut to the Brain Models Parkinson’s Disease. *Neuron* 103:627-641.e7. doi:10.1016/j.neuron.2019.05.035

Luk KC, Song C, O’Brien P, Stieber A, Branch JR, Brunden KR, Trojanowski JQ, Lee VM-Y. 2009. Exogenous α-synuclein fibrils seed the formation of Lewy body-like intracellular inclusions in cultured cells. *Proc National Acad Sci USA* 106:20051–20056. doi:10.1073/pnas.0908005106

Peng C, Gathagan RJ, Covell DJ, Medellin C, Stieber A, Robinson JL, Zhang B, Pitkin RM, Olufemi MF, Luk KC, Trojanowski JQ, Lee VM-Y. 2018. Cellular milieu imparts distinct pathological α-synuclein strains in α-synucleinopathies. *Nature* 557:558–563. doi:10.1038/s41586-018-0104-4

Rey NL, Steiner JA, Maroof N, Luk KC, Madaj Z, Trojanowski JQ, Lee VM-Y, Brundin P. 2016. Widespread transneuronal propagation of α-synucleinopathy triggered in olfactory bulb mimics prodromal Parkinson’s diseaseα-Synucleinopathy spreads across brain regions. *J Exp Medicine* 213:1759–1778. doi:10.1084/jem.20160368

Siksou L, Rostaing P, Lechaire J-P, Boudier T, Ohtsuka T, Fejtova A, Kao H-T, Greengard P, Gundelfinger ED, Triller A, Marty S. 2007. Three-Dimensional Architecture of Presynaptic Terminal Cytomatrix. *J Neurosci* 27:6868–6877. doi:10.1523/jneurosci.1773-07.2007

Thakur P, Breger LS, Lundblad M, Wan OW, Mattsson B, Luk KC, Lee VMY, Trojanowski JQ, Björklund A. 2017. Modeling Parkinson’s disease pathology by combination of fibril seeds and α-synuclein overexpression in the rat brain. *P Natl Acad Sci Usa* 114:E8284–E8293. doi:10.1073/pnas.1710442114

Vasili E, Dominguez-Meijide A, Flores-León M, Al-Azzani M, Kanellidi A, Melki R, Stefanis L, Outeiro TF. 2022. Endogenous Levels of Α-Synuclein Modulate Seeding and Aggregation in Cultured Cells. *Mol Neurobiol* 1–12. doi:10.1007/s12035-021-02713-2

Volpicelli-Daley LA, Luk KC, Patel TP, Tanik SA, Riddle DM, Stieber A, Meaney DF, Trojanowski JQ, Lee VM-Y. 2011. Exogenous α-Synuclein Fibrils Induce Lewy Body Pathology Leading to Synaptic Dysfunction and Neuron Death. *Neuron* 72:57–71. doi:10.1016/j.neuron.2011.08.033